# Dietary Fatty Acid Intake and the Colonic Gut Microbiota in Humans

**DOI:** 10.3390/nu14132722

**Published:** 2022-06-29

**Authors:** Anthony A. Xu, Luke K. Kennedy, Kristi Hoffman, Donna L. White, Fasiha Kanwal, Hashem B. El-Serag, Joseph F. Petrosino, Li Jiao

**Affiliations:** 1Department of Medicine, Baylor College of Medicine, Houston, TX 77030, USA; anthoney.xu@bcm.edu (A.A.X.); luke.kennedy@bcm.edu (L.K.K.); dwhite1@bcm.edu (D.L.W.); kanwal@bcm.edu (F.K.); hasheme@bcm.edu (H.B.E.-S.); 2The Alkek Center for Metagenomics and Microbiome Research, Department of Molecular Virology and Microbiology, Baylor College of Medicine, Houston, TX 77030, USA; kristi.hoffman@bcm.edu (K.H.); jpetrosi@bcm.edu (J.F.P.); 3Center for Innovations in Quality, Effectiveness and Safety, Michael E. DeBakey VA Medical Center, Houston, TX 77030, USA; 4Texas Medical Center Digestive Disease Center, Houston, TX 77030, USA; 5Section of Gastroenterology, Michael E. DeBakey VA Medical Center, Houston, TX 77030, USA

**Keywords:** diet, fat, microbiome, mucosa, human, epidemiology, *Sutterella*

## Abstract

A high-fat diet has been associated with systemic diseases in humans and alterations in gut microbiota in animal studies. However, the influence of dietary fatty acid intake on gut microbiota in humans has not been well studied. In this cross-sectional study, we examined the association between intake of total fatty acids (TFAs), saturated fatty acids (SFAs), trans fatty acids (TrFAs), monounsaturated fatty acids (MUFAs), polyunsaturated fatty acids (PUFAs), n3-FAs, and n6-FAs, and the community composition and structure of the adherent colonic gut microbiota. We obtained 97 colonic biopsies from 34 participants with endoscopically normal colons. Microbial DNA was used to sequence the 16S rRNA V4 region. The DADA2 and SILVA database were used for amplicon sequence variant assignment. Dietary data were collected using the Block food frequency questionnaire. The biodiversity and the relative abundance of the bacterial taxa by higher vs. lower fat intake were compared using the Mann–Whitney test followed by multivariable negative binomial regression model. False discovery rate–adjusted *p*-values (*q* value) < 0.05 indicated statistical significance. The beta diversity of gut bacteria differed significantly by intake of all types of fatty acids. The relative abundance of *Sutterella* was significantly higher with higher intake of TFAs, MUFAs, PUFAs, and n6-FAs. The relative abundance of *Tyzzerella* and *Fusobacterium* was significantly higher with higher intake of SFAs. *Tyzzerella* was also higher with higher intake of TrFA. These observations were confirmed by multivariate analyses. Dietary fat intake was associated with bacterial composition and structure. *Sutterella*, *Fusobacterium,* and *Tyzzerella* were associated with fatty acid intake.

## 1. Introduction

The associations between dietary fatty acid intake and cardiovascular diseases and cancer as well as all-cause mortality have been well investigated [1,2,3]. In general, trans-fatty acids (TrFAs) and saturated fatty acids (SAFs) are considered ‘unhealthy’, while monounsaturated fatty acids (MUFAs) and polyunsaturated fatty acids (PUFAs) are considered ‘healthy’ [3,4,5,6]. Accordingly, the American Heart Association and American College of Cardiology recommend avoiding TrFAs, limiting SFAs to <6% of total calories, and substituting those fats with MUFAs and PUFAs [7,8]. Long-chain PUFAs are further divided into n-3 and n-6 fatty acids (FAs), and these fatty acids are thought to modulate inflammation and provide cardiometabolic benefits [9]. Moreover, the underlying mechanisms of the associations between fat intake and health outcomes have been investigated. Some studies have shown that a high-fat diet (HFD) can induce Toll-like receptor 4 (TLR4) and trigger a downstream inflammatory response in mice [10,11]. Another study showed that mice on extended HFDs had an increase in inflammatory markers in the hypothalamus, possibly through the IKKβ/NF-κB pathway [12]. 

Alterations in gut microbiota have been associated with an increased risk of systemic diseases, including cardiovascular diseases [13] and cancers [14]. Diet has been shown to shape the gut microbiome [15,16]. Although dietary fatty acids are thought to be digested and absorbed in the small intestine [17,18], the influence of fatty acid intake on gut microbiota has been widely investigated in animal studies [19]. The studies in humans, however, have been limited. A recent systematic review of nine feces-based observational studies did not show consistent changes in taxonomic distribution based on dietary fat intake. A few interventional studies were relatively short in duration (3–6 weeks) [20]. Our previous analysis shows that a lower score for saturated fatty acid intake on the Healthy Eating Index (HEI) was associated with a higher relative abundance of *Fusobacterium* [21]. This finding suggests that examining the specific nutrients in association with gut microbiota would provide additional insight into the role of diet and microbiota in health.

Therefore, in the present analysis, following the lead of the holistic analysis on diet quality and gut microbiota, we investigated the association between the intake of different types of dietary fatty acids and the community composition and structure of the adherent colonic mucosal microbiota in humans. We hypothesized that there would be a positive association between SFA and TrFA intake and the abundance of potentially pro-inflammatory colonic microbiota and a positive association between MUFA and PUFA intake and the abundance of potentially anti-inflammatory colonic microbiota. Improved understanding of the association between dietary fat and gut microbiota may help refining the consensus and developing appropriate preventive and therapeutic strategies for diseases. 

## 2. Materials and Methods

### 2.1. Study Participants

Participants were prospectively and consecutively recruited from the endoscopy suite of the Michael E. DeBakey VA Medical Center (MEDVAMC) in Houston between 2013 and 2017. The methods, including comprehensive eligibility criteria, have been described previously [22]. Eligible participants had no cancer history or inflammatory bowel disease. They were not diagnosed with colorectal adenoma in the past three years and did not receive antibiotics in the prior three months. To be eligible for the present study, the participants had no abnormal findings during colonoscopy. 

### 2.2. Data Collection

Study participants were recruited 2 to 3 weeks before the colonoscopy, when the lifestyle and medical history were also evaluated using an interviewer-administered questionnaire. The interviewer measured the body weight and height of each participant after the interview. Participants self-administered the validated BLOCK Food Frequency Questionnaire (FFQ), which evaluates consumption of a wide variety of food groups over the prior 12 months [23]. The food list was developed from the NHANES 1999–2002 dietary recall data. The nutrient database was developed from the USDA Food and Nutrient Database for Dietary Study (version 1.0). Daily intake of food and nutrients, including TFAs, SFAs, TrFAs, MUFAs, PUFAs, n3-FAs, and n6-FAs, was calculated from the FFQ at the NutritionQuest. The daily intake value was energy-adjusted using the density method. The 2005 Healthy Eating Index (HEI) score, one of the measures of dietary quality, was calculated [24].

### 2.3. Tissue Collection and DNA Extraction

Participants were advised to stop taking aspirin, anti-inflammatory drugs, blood thinners, iron, or vitamins with iron 7 days prior to the colonoscopy and stop diabetic medication one day before the procedure. On the day of colonoscopy, the endoscopists obtained biopsies from each colonic segment (cecum; ascending colon; transverse, descending, and sigmoid colon; and rectum) when the patients were found to have grossly normal-appearing colon mucosa. The acquired biopsies were stored at −80 °C within 15 min of sample collection [22].

We enrolled 612 eligible study participants in the study. Among them, 174 were found to have a normal colon, and 134 of them consented to provide colonic mucosal biopsy. Samples from 69 participants were sent for microbiota profiling. Among them, a total of 40 participants responded to the FFQ. Five study participants who had self-reported energy consumption <800 or >5000 kcal per day were excluded from the analysis. Therefore, the raw sequencing data of 99 mucosal samples from 35 participants were generated (Appendix A). The study protocol was approved by the Institutional Review Board of Baylor College of Medicine (BCM) and MEDVAMC. All study participants provided informed consent to participate.

### 2.4. 16S rRNA Sequencing

The sequence analyses were performed at the Alkek Center for Metagenomics and Microbiome Research (CMMR) at BCM. Bacterial genomic DNA was extracted from the biopsies using the MO BIO PowerLyzer UltraClean Tissue & Cell DNA Isolation Kits (MO BIO Laboratories, Clardad, CA, USA). All DNA samples were stored at −80 °C until further analysis. 

The 16S rRNA gene sequencing methods were adapted from the published methods [25,26,27]. The 16S rRNA hypervariable region 4 (V4) was amplified by PCR using the barcoded Illumina adaptor-containing primers 515F and 806R and sequenced on the MiSeq platform (Illumina, San Diego, CA, USA) using the 2 × 250 bp paired-end protocol. The primers used for amplification contained adapters for the MiSeq sequencing and single-index barcodes so that the PCR products could be pooled and sequenced directly [27].

### 2.5. Bioinformatics and Taxonomic Assignment

We used the CMMR bioinformatics pipeline for data analysis. The reads were merged using USEARCH v7.0.1090 [28]. A quality filter was applied to the resulting merged reads, and those containing above 0.5% expected errors were discarded. We used the Divisive Amplicon Denoising Algorithm 2 (DADA2) v1.10.1 package in R v3.3.3 to classify the bacteria using the Amplicon Sequence Variant (ASV). The ASVs were mapped to the SILVA v128 to determine taxonomies [29,30]. A rarefied ASV table was used for downstream analyses of biodiversity and phylogenetic trends using the Agile Toolkit for Incisive Microbial Analyses (ATIMA) [31]. A rarefaction curve, using the factor 4356, was constructed using the sequence data for each sample to ensure that we sampled the majority of its microbial diversity. After rarefaction, two mucosal samples from one individual had poor sequencing data were eliminated. We were left with 97 mucosal samples from 34 participants for the final analysis (Appendix A). 

### 2.6. Statistical Analysis

The bacterial alpha-diversity, beta-diversity, and the relative abundance of bacterial taxa (mainly at the phylum, family, and genus level) were compared based on higher vs. lower intake of dietary TFAs, SFAs, TrFAs, MUFAs, PUFAs, n3-FAs, and n6-FAs. Higher vs. lower intake was dichotomized using the median intake for each nutrient in 34 participants. Sociodemographic and clinical characteristics of the participants were compared according to SFA intake using the Student’s *t* test or Fisher’s exact test. We examined participant characteristics by SFA because we observed the score for SFAs was associated with gut microbiota in our previous study [21]. Permutational multivariate analysis of variance (PERMANOVA) was used to evaluate beta-diversity using the weighted Bray–Curtis as the distance matrix. The principal coordinate analysis (PCoA) plots [32] were constructed to visualize the dissimilarity of microbial community composition by fatty acid intake. The relative abundance of bacterial taxa was compared between higher vs. lower intake groups using the Mann–Whitney test. 

We used multivariable negative binomial regression models for panel data to examine the association between fatty acid intake and bacterial counts, adjusting for age (continuous), ethnicity (non-Hispanic White, African American, and Hispanic), body mass index (BMI, continuous), smoking status (never, former, and current smokers), alcohol use (never, former, and current drinkers), HEI score (continuous), hypertension (yes vs. no), diabetes (yes vs. no), and colon segment (cecum; ascending, transverse, descending, and sigmoid colon; and rectum). Each participant was treated as a panel because multiple biopsies were taken from some of the participants. The incidence rate ratio (IRR) and its 95% confidence interval (CI) of having a non-zero bacterial count in higher vs. lower intake groups were estimated. Because dietary fat intake also affects the amount of daily calories from protein and carbohydrates that could impact the gut microbiota [33,34], we also included total protein and carbohydrate intake in the models. 

We used the STATA 16.0 (Stata Corp LLC, College Station, TX, USA) and the R program for data analysis. A *p* value < 0.05 indicated statistical significance. In the microbiota analysis, all *p*-values were adjusted for multiple comparisons using the false discovery rate (FDR) algorithm [35]. FDR-adjusted *p*-values (*q* values) < 0.05 indicated statistical significance.

## 3. Results

### 3.1. General Characteristics of Study Participants

Our study consisted of 33 men and 1 woman, aged 51 to 71 years old. The majority of participants were non-Hispanic White (71%) and men (97%). Table 1 shows that there were no statistically significant differences in the distribution of demographics and lifestyle factors based on intake of SFAs. However, those who had a lower intake of SFAs had a significantly higher HEI score and higher intake of total carbohydrates. 

In our study, the mean daily intakes of TFA, SFA, TrFA, MUFA, PUFA, n3-FA, and n6-FA were 42.2, 13.1, 1.39, 16.5, 9.31, 0.89, and 8.04 g/1000 kcal, respectively.

### 3.2. Biodiversity

The microbial alpha diversity only differed by TrFA intake. Compared to individuals with lower TrFA intake, the gut microbiota of individuals with higher intake of TrFAs had a lower alpha diversity (*q* value = 0.02 for the Shannon index). There were significant differences in beta diversity by intake of all types of fatty acids (Figure 1).

Figure 1 Principal coordinate analysis (PCoA) with weighted Bray–Curtis dissimilarity shows that the bacterial beta diversity differed significantly between higher vs. lower intake of multiple fatty acids (*p* values ≤ 0.006, PERMANOVA test). The centroids of the two groups did not overlap. The fraction of diversity captured by the coordinate is shown as a percentage in the corresponding axis. PC1 and PC2 represent the top two principal coordinates that capture most of the diversity.

### 3.3. Taxonomy

Seven phyla with relative abundance of greater than 0.5% were identified in our study sample. The relative abundance was 43.0% for Firmicutes, 35.6% for Bacteroidota, 9.00% for Proteobacteria, 3.18% for Verrucomicrobiota, 1.53% for both Fusobacteriota and Desulfobacterota, and 0.88% for Actinobacteriota. At the phylum level, Table 2 shows the relative abundance of Desulfobacteria was significantly higher with higher intake of TFAs, MUFAs, PUFAs, and n6-FAs. The relative abundance of Fusobacteria was significantly higher with higher intake of SFAs and MUFAs. At the family level, Appendix A shows the relative abundance of Sutterellaceae and Desulfovibrionaceae was higher with higher intake of TFAs, MUFAs, PUFAs, and n6-FAs. The relative abundance of Fusobacteriaceae was higher with higher intake of SFAs and MUFAs. The relative abundance of Acidaminococcaceae was higher with higher intake of TFAs, PUFAs, n3-FAs, and n6-FAs. The relative abundance of Christensenellaceae was lower with higher intake of TFAs, TrFA, PUFAs, and n6-FAs. Other families, such as Prevotellaceae, Anaerovacaceae, Erysipelotrichaceae, and Ruminococcaceae, also differed significantly by fatty acid intake (Appendix A).

Table 3 shows that the relative abundance of the bacterial genera differed significantly by intake of multiple types of dietary fatty acids. The relative abundance of *Sutterella* was higher with the higher intake of TFAs, MUFAs, PUFAs, n3-FAs, and n6-FAs. The relative abundance of *Tyzzerella*, *Butyricimonas*, and *Bifidobacterium* was higher with the higher intake of SFAs and TrFA. The relative abundance of *Fusobacterium* was higher with the higher intake of SFAs and MUFAs. The relative abundance of *Faecalibacterium* was higher with the higher intake of n3-FAs. The relative abundance of *Acidaminococcus* was higher with the higher intake of TFAs, PUFAs, n3-FAs, and n6-FAs.

As seen in Table 4, the multivariable analysis confirmed the associations between *Sutterella* and TFAs, MUFAs, PUFAs, and n6-FAs, which were independent of dietary quality because the adjustment of HEI in the model did not change the IRRs. The incidence rate ratio of having non-zero *Sutterella* count in participants with higher intake of TFAs was 24% higher than the participants with a lower intake of TFAs.

The association between SFA and TrFA intake and *Tyzzerella* was borderline significant after adjusting for dietary quality. The association between *Fusobacterium* and SFAs was attenuated when HEI was adjusted in the model. All other differences in bacterial relative abundance by fat intake were not statistically significant in the multivariable analyses. Adjustment of protein and carbohydrate did not change the estimate significantly after dietary quality was included in the model (data not shown). 

## 4. Discussion

Our study found that intake of dietary fatty acids significantly associated with the community composition and structure of the adherent gut microbiota in the colon. Members of the Sutterellaceae, Desulfovibrionaceae, Erysipelotrichaceae, Fusobacteriaceae, and Christensenellaceae families differed by fatty acid intake. The relative abundance of *Sutterella, Tyzzerella, and Fusobacterium* differed significantly by fatty acid intake. 

Experimental studies have shown that increased dietary fat intake is related to a lower gut microbial alpha diversity [36,37,38]. A systematic review showed that a higher intake of SFAs was associated with a decrease in microbial richness and diversity in humans [20]. However, our study did not show a significant difference in alpha diversity based on fatty acid intake, except for TrFAs. A higher intake of TrFAs was related to a lower Shannon index. On the other hand, the significant differences in the beta diversity (community composition) of gut microbiota were found for all types of fatty acids examined. A French study using fecal samples showed a significant difference in beta diversity based on a diet that includes various fatty foods, although the type of fat was not specified in the study [39]. In summary, our observations suggest that the intake of fatty acids could be associated with the community composition of the gut microbiota in the colon in humans. The lower bacterial richness related to a higher intake of TrFAs may partially explain its harmful influence on health. 

A higher intake of TFAs, MUFAs, PUFAs, and n6-FAs was associated with a higher relative abundance of *Sutterella* compared to a lower intake. Higher *Sutterella* with higher intake of n3-FAs was not confirmed by the multivariable analyses. The depletion of *Sutterella* has been observed in children with autism [40] and with a lower quality of life in patients with Crohn’s disease [41]. A diet of bodybuilders (high protein, high fat, low carbohydrate, and low fiber) has been correlated with increased *Sutterella* [42]. We previously showed that *Sutterella* was depleted in people who slept for less than 6 hours per day compared to people slept for 6–8 h per day [43]. One animal study showed that *Sutterella* may have beneficial effects on glycometabolism in diabetic mice fed with an HFD [44]. On the other hand, increased relative abundance of *Sutterella* has been associated with the increased risk of metabolic syndrome, Down’s syndrome, autism, and inflammatory bowel disease [45,46,47,48,49]. Despite conflicting evidence on the beneficial vs. detrimental influence of *Sutterella* on health outcomes, all studies have shown that diet including fatty acids can modulate *Sutterella*. The role of *Sutterella* in metabolism, neurological symptoms, and inflammation through diet warrants further investigation. 

A higher intake of SFAs was associated with a higher relative abundance of *Fusobacterium*. Our observation was in agreement with another study that showed an increase in *Fusobacterium* when participants switched from a high fiber diet to an HFD [50]. *Fusobacterium,* a Gram-negative bacterium with lipopolysaccharides, has been shown to accelerate atherosclerosis in rabbits [50] and has been associated with colorectal cancer, atherosclerosis, and a lower HEI score in humans [21,51,52]. This pro-inflammatory property is likely from its ability to modulate the immune system. One observational study suggested that *Fusobacterium*, which was found in much higher levels in cancerous tissue as compared to control tissue, increases cancer risk by increasing inflammatory mediators through a possible miRNA-mediated activation of *TLR2*/*TLR4* [53]. However, the positive association between *Fusobacterium* and SFA intake was attenuated when dietary quality was included in the model. The association between *Fusobacterium* and MUFA intake was not shown in multivariable analysis.

A higher intake of SFAs and TrFAs was associated with a higher relative abundance of *Tyzzerella*. These associations persisted in the multivariable analyses. *Tyzzerella* is a member of the Lachnospiraceae family and has been associated with a higher risk of cardiovascular diseases and a lower HEI score [21,54]. Patients with Crohn’s disease were reported to have significantly more abundant *Tyzzerella* than patients without Crohn’s disease [55]. One study showed that *Tyzzerella* was one of the bacteria that significantly decreased in hyperlipidemic rats after treatment with simvastatin [56]. However, the exact mechanism by which *Tyzzerella* may mediate the potential harmful effects of fatty acid intake in cardiovascular or other diseases is unclear. 

A higher intake of TFAs, PUFAs, n3-FAs, and n6-FAs was associated with an increased relative abundance of *Acidaminococcus*. *Acidaminococcus* is a genus in the Firmicutes phylum and Negativicutes class. *Acidaminococcus* is a glutamate-fermenting microbe [57] and glutamate has been shown to provide oxidative fuel for the intestinal epithelium and play an important role in maintaining normal gut barrier function [58,59]. PUFAs have been shown to decrease the incidence of coronary heart disease and have a beneficial effect on glycemic control and cancer [60,61]. The role of PUFAs in supporting colonic epithelial barrier integrity has also been reported [62]. Previous studies have suggested that n3-FAs could increase the relative abundance of *Faecalibacteria* and short-chain-fatty acid-generating bacteria in animal models or in humans [63]. However, in the present analysis, we did not observe a significant association between n3-FA intake and gut microbiota in multivariable analysis. The Western type of diet has a higher n6-FA:n3-FA ratio [64]. Our study participants had much lower n3-FA intake than n6-FA intake, with an average ratio of 9.0. Large studies in other study populations are warranted to further investigate the multivariable association between *Acidaminococcus* and diet as well as n3-FA and gut microbiota. 

A unique feature of our study was that we examined the adherent mucosal bacteria in association with intake of different types of fatty acids in humans. We included other lifestyle factors and dietary quality as the confounding factors in the multivariable models. The association between *Fusobacterium* and SFAs was attenuated when HEI was adjusted. Our study had several limitations. First, the study finding was mostly based on male veterans, which limited its generalizability to other populations. Second, we used the arbitrary median as the divider between higher vs. lower dietary intake in our analysis. Third, because we relied on patients’ subjective reports on food consumed in the previous year, information bias cannot be excluded. Fourth, the 16S rRNA taxonomic survey has limited sequencing resolution and could not identify the bacterial species in association with fat intake. There were abundant unnamed bacteria that showed significant differences in relative abundance by fatty acid intake. Future studies should identify those bacteria genus or species that could have biological consequence. Fifth, we could not completely rule out the influence of anti-diabetic and anti-hypertensive medications on the gut microbial profile because participants were allowed to use these medications up to the date of the procedure. Future studies should investigate the interaction between diet, medication use, and gut microbiota. Last, our study sample size was small, but if enlarged, would have given us more powerful statistical analyses.

## 5. Conclusions

Various types of fatty acids may influence gut microbiota differentially and explain their respective effects on systemic diseases. The genera that are most associated with intake of fatty acids include *Sutterella, Tyzzerella,* and *Fusobacterium.* Our findings were consistent with our hypothesis that TrFAs and SFAs were positively associated with *Tyzzerella* and SFAs were positively associated with pro-inflammatory *Fusobacterium*. However, the findings on *Sutterella* and health outcomes are conflicting. Identifying bacterial species that are associated with fatty acid intake and immune and inflammatory modulation may provide novel preventive and treatment modalities for more tailored health management. 

## Figures and Tables

**Figure 1 nutrients-14-02722-f001:**
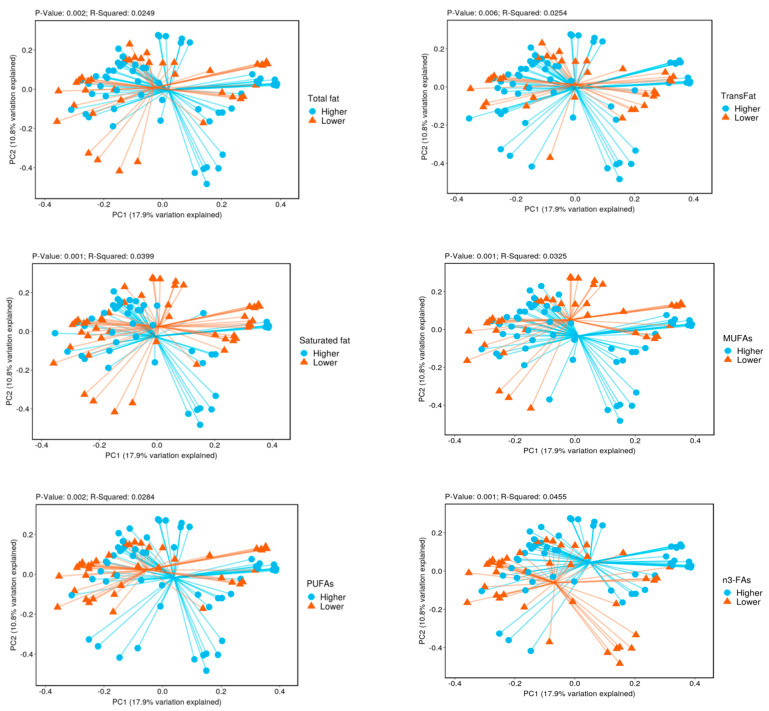
Beta diversity by lower vs. higher intake of fatty acids.

**Table 1 nutrients-14-02722-t001:** Patient characteristics by higher vs. lower saturated fatty acid intake.

Characteristics Mean ± Standard Deviation or *n* (%)	Lower Intake *n* = 17	Higher Intake *n* = 17	*p* Value ^a^
Age (years)	61.1 ± 6.0	62.9 ± 5.1	0.34
Men, *n* (%)	16 (95%)	17 (100%)	0.31
Racial group			
White, *n* (%)	11 (65%)	13 (76%)	0.66
African American, *n* (%)	4 (23%)	2 (12%)	
Hispanic, *n* (%)	2 (12%)	2 (12%)	
Body mass index (kg/m^2^)	32.4 ± 7.4	35.4 ± 5.2	0.18
Ever smokers, *n* (%)	10 (59%)	11 (65%)	0.72
Current alcohol use, *n* (%)	8 (47%)	7 (41%)	0.46
Hypertension, yes (%)	11 (65%)	14 (82%)	0.24
Type 2 diabetes, yes (%)	7 (41)	10 (59)	0.30
Healthy Eating Index	64.1 ± 8.7	57.8 ± 8.3	0.04
Saturated fat (grams/1000 kcal/day)	10.5 ± 0.98	15.7 ± 2.16	<0.0001
Total carbohydrates (grams/1000 kcal/day)	123 ± 21	103 ±14	0.004
Protein (grams/1000 kcal/day)	35.7 ± 8.1	40.5 ± 5.7	0.05

^a^*p* value for two-sample *t* test or Fisher’s exact test.

**Table 2 nutrients-14-02722-t002:** Relative abundance of bacterial phylum by fatty acid intake.

Type of Fatty Acid	Phylum	Relative Abundance (%)		*q* Value ^a^
		Lower Intake	Higher Intake	
TFAs	*Desulfobacterota*	1.05	1.84	0.005
MUFAs	*Desulfobacterota*	1.10	1.82	0.029
	*Fusobacteria*	0.84	1.99	0.008
PUFAs	*Desulfobacterota*	1.04	1.89	0.013
n6-FA	*Desulfobacterota*	1.04	1.89	0.013
SFAs	*Fusobacteria*	0.70	2.34	0.008

MUFAs: monounsaturated fatty acids; PUFAs: polyunsaturated fatty acids; SFAs: saturated fatty acids; TFAs: total fatty acids. ^a^ *q* value for the Mann–Whitney test.

**Table 3 nutrients-14-02722-t003:** Relative abundance of bacterial genus by fatty acid intake.

Type Of Fatty Acid	Genus	Relative Abundance (%)		*q* Value ^a^
		Lower Intake	Higher Intake	
TFAs	*Sutterella*	0.57	2.41	<0.0001
	*Acidaminococcus*	0.03	0.24	0.04
MUFAs	*Lachnoclostridium*	0.87	1.57	0.002
	*Sutterella*	0.80	2.29	0.002
	*Founierella*	0.02	0.13	0.035
	*Fusobacterium*	0.82	1.97	0.036
	*Intestinibacter*	0.15	0.17	0.041
PUFAs	*Anaerostipes*	0.91	0.12	0.0006
	*Sutterella*	0.66	2.44	0.0006
	*Acidaminococcus*	0.03	0.26	0.01
	*Bilophila*	0.50	0.99	0.02
	*Colidextribactor*	0.14	0.33	0.02
	*Prevotella*	1.50	3.39	0.02
n3-FAs	*Alloprevotella*	0.60	0	0.004
	*Faecalibacterium*	4.34	9.92	0.004
	*Subdoligranulum*	0.21	0.73	0.004
	*Acidaminococcus*	0.03	0.26	0.007
	*Sutterella*	0.85	2.30	0.026
	*Phascolarctobacterium*	0.44	0.78	0.03
	*Tyzzerella*	0.45	0.14	0.03
n6-FAs	*Anaerostipes*	0.91	0.12	<0.001
	*Sutterella*	0.66	2.44	<0.001
	*Acidaminococcus*	0.03	0.26	0.007
	*Bilophila*	0.50	0.99	0.007
	*Colidextribacter*	0.14	0.33	0.014
	*Prevotella*	1.51	3.39	0.014
SFAs	*Tyzzerella*	0.11	0.43	0.0004
	*Negativibacilus*	0.18	0.04	0.003
	*Butyricimonas*	0.15	0.20	0.032
	*Fusobacterium*	0.66	2.37	0.030
	*Bifidobacterium*	0.14	0.41	0.031
	*Intestinibacter*	0.26	0.06	0.031
	*Veillonella*	0.14	0.42	0.031
TrFAs	*Bifidobacterium*	0.19	0.36	0.008
	*Tyzzerella*	0.10	0.34	0.008
	*Butyricicoccus*	0.13	0.34	0.028

MUFAs: monounsaturated fatty acids; PUFAs: polyunsaturated fatty acids; SFAs: saturated fatty acids; TFAs: total fatty acids. TrFAs: trans fatty acids. ^a^ *q* value for the Mann-Whitney test.

**Table 4 nutrients-14-02722-t004:** The incidence rate ratio (IRR) of having non-zero bacterial count by fatty acid intake.

Genus	Type of Fatty Acid	Count		IRR (95% CI) ^b^	IRR (95% CI) ^c^	IRR (95% CI) ^d^
		Lower Higher Intake ^a^ Intake				
		Median count				
*Sutterella*	TFAs	0	18	1.25 (1.15–1.37)	1.24 (1.12–1.37)	1.24 (1.12–1.37)
*Sutterella*	MUFAs	0	17	1.67 (1.35–2.12)	1.65 (1.32–2.08)	1.65 (1.31–2.07)
*Sutterella*	PUFAs	0	17	1.24 (1.04–1.47)	1.19 (0.95–1.48)	1.48 (1.12–1.94)
*Sutterella*	n6-FAs	0	17	1.30 (1.05–1.60)	1.23 (0.94–1.61)	1.55 (1.11–2.16)
*Tyzzerella*	SFAs	0	14.5	1.18 (0.76–1.83)	2.04 (1.19–3.48)	1.66 (1.00–2.76)
*Fusobacterium*	SFAs	0	5	1.21 (1.02–1.43)	1.38 (1.10–1.71)	1.11 (0.83–1.48)
*Tyzzerella*	TrFAs	0	0	1.52 (0.17–13.3)	6.62 (1.01–43.0)	6.61(1.02–43.0)

CI: confidence interval; IRR: incidence rate ratio; MUFAs: monounsaturated fatty acids; PUFAs: polyunsaturated fatty acids; SFAs: saturated fatty acids; TFAs: total fatty acids. TrFAs: trans fatty acids. ^a^ Lower intake of fatty acids was the reference group in the negative binomial regression model for panel data. The lower intake was defined as lower than median intake in 34 participants. ^b^ The model was adjusted for age. ^c^ The model was adjusted for age, ethnicity, BMI, alcohol use, smoking, hypertension, diabetes, and segment. ^d^ The model was adjusted for HEI score in addition.

## Data Availability

Data available on request due to local policy on privacy.

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
