# Peer review of "Dietary Fatty Acid Intake and the Colonic Gut Microbiota in Humans"

_nutrients, 2022, doi:10.3390/nu14132722_

Round 1

Reviewer 1 Report

General comments:

This observational study examined associations of intake of fatty acids (saturated, trans, monounsaturated, polyunsaturated, n3- and n6- fatty acids) with the structure and composition of the adherent colonic gut microbiota. Feces samples are more commonly used for gut microbiota analysis. It is of interest to mention how the composition of the microbiota obtained by colonic biopsies corresponds to that assessed in feces. To address the relationships “diet – chronic diseases” in humans, it has been preferred an approach of the whole diet instead of isolated nutrients. Additionally, this reviewer was wondering how to interpret an association of a nutrient with relative abundance of certain commensal bacteria among trillions that inhabit the human gut.

Specific comments:

·      Introduction

The background to justify the present study was adequate since underlying mechanisms of the associations between fatty acids intake and health outcomes should be mediated by the gut microbiota. However, it is suggested to add that a healthy diet should be balanced in omega-6 and omega-3 PUFA and that today’s Western diets which promote diseases have a high omega-6 to omega-3 ratio.

In the third paragraph, authors hypothesized that there would be a positive association of saturated and trans fatty acids intakes with a potentially pathogenic microbiota, and a positive association of MUFA and PUFA intakes with potentially anti-inflammatory microbiota. Please, consider changing the adjective “pathogenic” since most bacteria investigated in this study are normally found in human intestine. Overrepresentation of some bacteria has been associated with a pro-inflammatory condition.

·      Materials and Methods

Eligibility criteria were adequate (normal colonoscopy, no antibiotics for the last 3 months.

Food frequency questionnaire at the NutritionQuest was an appropriate tool to calculate macronutrients including the types of fatty acids. However, there is a more recent version of the 1995 Health Eating Index (HEI). Please, consider using the 2005 HEI.

Regarding the advice to stop medication prior to colonoscopy, just one day for metformin seems insufficient to neutralize its effects on microbiota composition. This antidiabetic agent has been associated with increased relative abundance of mucin-degrading and SCFA-producing bacteria. Please, comment.

This reviewer would suggest including a flowchart of the final sample, starting with the 174 individuals who had a normal colon. A total of 97 mucosal samples from 34 participants (1 woman) was studied.

Regarding statistical analysis of the microbiota data, only two categories were created (lower or higher intake) due to the sample size, and their relative abundances of bacteria were compared by Mann-Whitney. This reviewer was wondering whether other tests would be more appropriate for comparison of relative abundance between subgroups. Have the authors consider using the DESeq2 for differential abundance, for instance?

·      Results

In table 1, participants were stratified by the intake of saturated fatty acids instead of total fat. Is there a reason for that? The title of figure 1 is missing. In the topic Taxonomy, this reviewer suggests describing the relative abundance of the most important phyla (Firmicutes, Bacteroidetes, Actinobacteria and Proteobacteria) for the entire sample. Table 2 compares the relative abundances of two phyla (Desulfobacteria and Fusobacteria) and table 3 of numerous genera between subgroups of individuals stratified by fatty acids intake. The selected genera have, in general, low representativeness in the microbiota composition. It is important to add footnotes in these tables informing which test used for comparison.  

·      Discussion

The first paragraph states that “(…) bacterial genera that were mostly influenced by fatty acids intake. Considering the cross-sectional design is not appropriate to infer causality, it is recommended to avoid the verb “influence”. The present findings were compared to previous reports and some underlying mechanisms of the associations were commented.

Several limitations were mentioned but other aspects should be included in the discussion section. To address the relationships “diet – chronic diseases” in humans, it has been preferred an approach of the whole diet instead of isolated nutrients. Additionally, this reviewer was wondering how to interpret an association of a nutrient with relative abundance of certain commensal bacteria among trillions that inhabit the human gut.

Author Response

Manuscript ID 1760154: Nutrients
Type of manuscript: Article
Title: Dietary fatty acids intake and the colonic gut microbiota in humans

Dear editors:

We appreciate reviewers’ insightful comments on our manuscript. Please see the point-by-point response below. Please see the response in brown character. Thank you!

General comments:

This observational study examined associations of intake of fatty acids (saturated, trans, monounsaturated, polyunsaturated, n3- and n6- fatty acids) with the structure and composition of the adherent colonic gut microbiota. Feces samples are more commonly used for gut microbiota analysis.

It is of interest to mention how the composition of the microbiota obtained by colonic biopsies corresponds to that assessed in feces.

In a previously published study, we compared the rectal microbiota and fecal microbiota.

Jiao, L., Kourkoumpetis, T., Hutchinson, D. et al. Spatial Characteristics of Colonic Mucosa-Associated Gut Microbiota in Humans. Microb Ecol 83, 811–821 (2022). https://doi.org/10.1007/s00248-021-01789-6

In brief, the fecal bacterial composition is different from rectal mucosal microbiota.

To address the relationships “diet – chronic diseases” in humans, it has been preferred an approach of the whole diet instead of isolated nutrients. Additionally, this reviewer was wondering how to interpret an association of a nutrient with relative abundance of certain commensal bacteria among trillions that inhabit the human gut.

Thank you for the comments. When we used the whole diet (healthy eating index 2005) approach (PMCID: 31291462), we found that Fusobacterium was higher among those who had a lower score for saturated fatty acid intake. This finding motivated us to further investigate the association between dietary fatty acids and gut microbiota. We hypothesized that certain bacteria enrich when specific nutrient is abundant. The research also aimed to investigate the relevance of gut microbiota in light of numerous studies showing the association between specific fatty acids and health outcomes. More comments are provided below.

Specific comments:

Introduction

The background to justify the present study was adequate since underlying mechanisms of the associations between fatty acids intake and health outcomes should be mediated by the gut microbiota.

Q1. However, it is suggested to add that a healthy diet should be balanced in omega-6 and omega-3 PUFA and that today’s Western diets which promote diseases have a high omega-6 to omega-3 ratio.

Response: Thank you for raising this point. We included the average intake of different type of fatty acids in the Results. Please see the text below Table 1 on Page 5. The ratio of omega-6 to omega-3 was 9.0 in our study.

It reads “In our study, the mean daily intake of TFA, SFA, TrFA, MUFA, PUFA, n3-FA, and n6-FA was 42.2, 13.1, 1.39, 16.5, 9.31, 0.89, and 8.04 gram/1000 Kcal, respectively.  “

We further discussed the findings on n3-FA and n6-FA in the revision. It reads ”In the present analysis, we did not observe the association between n3-FA intake and gut microbiota in multivariable analysis. The Western-type of diet has a higher n6-FA:n3-FA ratio. Our study participants had much lower n3-FA intake than n6-FA intake, with an average ratio of 9.0. Large studies in other study populations are warranted to further investigate the multivariable association between Acidaminococcus and diet, as well as n3-FA and gut microbiota. Please see page 10, the first paragraph. Accordingly, references 66 and 67 were added in the revision.

Q2. In the third paragraph, authors hypothesized that there would be a positive association of saturated and trans fatty acids intakes with a potentially pathogenic microbiota, and a positive association of MUFA and PUFA intakes with potentially anti-inflammatory microbiota. Please, consider changing the adjective “pathogenic” since most bacteria investigated in this study are normally found in human intestine. Overrepresentation of some bacteria has been associated with a pro-inflammatory condition.

Response: Fully agreed. We changed the term to “potentially pro-inflammatory”. We hope it is more acceptable than “pathogenic”. Please let us know if further change is needed. Please see the 3rd paragraph on page 2.

Materials and Methods

Q3. Eligibility criteria were adequate (normal colonoscopy, no antibiotics for the last 3 months).

Food frequency questionnaire at the NutritionQuest was an appropriate tool to calculate macronutrients including the types of fatty acids. However, there is a more recent version of the 1995 Health Eating Index (HEI). Please, consider using the 2005 HEI.

Response: Thank you so much for the comment. We did use the 2005-HEI as we stated in our previous publications (reference 21). The reference was also updated. Please see line 1 on page 3.  

Q4. Regarding the advice to stop medication prior to colonoscopy, just one day for metformin seems insufficient to neutralize its effects on microbiota composition. This antidiabetic agent has been associated with increased relative abundance of mucin-degrading and SCFA-producing bacteria. Please, comment.

Response: Thank you for the question. We collected the complete medication use data from 28 participants. We did not find current use of metformin was related to a higher relative abundance of mucin-degrading bacteria (such as Akkermansia) and SCFA-producing bacteria (such as Faecalibacterium) in human colonic mucosa. Metformin use was related to a higher relative abundance of Escherichia in our study samples. However, Escherichia was not one of the bacteria to be associated with fatty acid intake in our study. In addition, it is noted that the bacteria identified in our study (Sutterella, Fusobacteria, and Tyzzerella) are not mucin-degrading and SCFA-producing.

Nevertheless, we included this statement in Limitation “We could not completely rule out the influence of anti-diabetic and anti-hypertensive medication on the gut microbiota profile because patients were allowed to use these medications up to the date of procedure. Future studies should investigate the interaction between diet, commonly used medications, and gut microbiota.” Please see Page 10, the last sentence above Conclusion.

Q5. This reviewer would suggest including a flowchart of the final sample, starting with the 174 individuals who had a normal colon. A total of 97 mucosal samples from 34 participants (1 woman) was studied.

Response: Thank you for the comment. A flow chart is included as Supplemental Figure 1 in this revision.

Q6: Regarding statistical analysis of the microbiota data, only two categories were created (lower or higher intake) due to the sample size, and their relative abundances of bacteria were compared by Mann-Whitney. This reviewer was wondering whether other tests would be more appropriate for comparison of relative abundance between subgroups. Have the authors consider using the DESeq2 for differential abundance, for instance?

Response: We thank the reviewer for the suggestion and admit to considering many different methods for analyzing our data. However, data analysis is a very dynamic aspect of the microbiome field, and while more complex methods such as DESeq2, ANCOM, and others are being developed or co-opted for such data, their reported benefits have thus far been minimal and highly dependent on dataset characteristics. For this reason, we chose the simple statistic of Mann-Whitney, which is not only easy to implement but also shows one of the best balances between sensitivity and false discovery rate. We cited to relevant references here.

Weiss, S., Xu, Z.Z., Peddada, S. et al. Normalization and microbial differential abundance strategies depend upon data characteristics. Microbiome 5, 27 (2017). https://doi.org/10.1186/s40168-017-0237-y 

Stijn Hawinkel, Federico Mattiello, Luc Bijnens, Olivier Thas, A broken promise: microbiome differential abundance methods do not control the false discovery rate, Briefings in Bioinformatics, Volume 20, Issue 1, January 2019, Pages 210–221, https://doi.org/10.1093/bib/bbx104 

Results

Q7. In table 1, participants were stratified by the intake of saturated fatty acids instead of total fat. Is there a reason for that? The title of figure 1 is missing.

Response: We apologize for not providing the Figure title. Both Figure title and the legend have been added to the revision. We chose to present the characteristics by saturated fat because we found saturated fatty acids as an HEI score component was associated with Fusobacterium in our previous analysis (PMCID: 31291462). We included this point in line 7 of section 2.6 on Page 3.

It reads “We examined participants’ characteristics by SFAs because we observed the score of SFAs was associated with gut microbiota in our previous study [21].”

Q8. In the topic Taxonomy, this reviewer suggests describing the relative abundance of the most important phyla (Firmicutes, Bacteroidetes, Actinobacteria and Proteobacteria) for the entire sample.

Response: We agreed that we should have provided this basic information to readers. The relative abundance of these important phyla was added to the manuscript as the text.  

It reads “Seven phyla with relative abundance of greater than 0.5% were identified in our study sample. The relative abundance of was 43.0% for Firmicutes, 35.6% for Bacteroidota, 9.00% for Proteobacteria, 3.18% for Verrucomicrobiota, 1.53% for both Fusobacteriota and Desulfobacterota, and 0.88% for Actinobacteriota”. Please see paragraph 2 on Page 6.

Q9, Table 2 compares the relative abundances of two phyla (Desulfobacteria and Fusobacteria) and table 3 of numerous genera between subgroups of individuals stratified by fatty acids intake. The selected genera have, in general, low representativeness in the microbiota composition. It is important to add footnotes in these tables informing which test used for comparison.  

Response: The test has been added as the footnote to each Table. Thank you.

Discussion

Q10. The first paragraph states that “(…) bacterial genera that were mostly influenced by fatty acids intake. Considering the cross-sectional design is not appropriate to infer causality, it is recommended to avoid the verb “influence”. The present findings were compared to previous reports and some underlying mechanisms of the associations were commented.

Response: We concur with the reviewer. Now the sentence is read “The relative abundance of Sutterella, Tyzzerella, and Fusobacterium differed significantly by fatty acids intake”. Please see first paragraph on Page 9. The corresponding sentence in Abstract has also been updated in the revision. It reads “Sutterella, Fusobacterium and Tyzzerella were significantly associated with fatty acid intake.”

Q11. Several limitations were mentioned but other aspects should be included in the discussion section. To address the relationships “diet – chronic diseases” in humans, it has been preferred an approach of the whole diet instead of isolated nutrients. Additionally, this reviewer was wondering how to interpret an association of a nutrient with relative abundance of certain commensal bacteria among trillions that inhabit the human gut.

Response: Thank you for the insightful comments. As noted above, this analysis was led by our previous findings on HEI and gut microbiome where we observed a lower score for saturated fat was associated with a higher relative abundance of Fusobacterium. The present analysis confirmed the previous findings and further identified Sutterella and Tyzzerella to differ by fatty acid intake.

In addition, numerous studies have found the association between specific nutrients (such as fatty acids, B vitamins, D vitamins, and etc.) and various outcomes. Whether gut microbiota plays any role in such association is currently unknown. We hypothesized that the bacteria were fed and enriched by specific nutrients and thereby exert functions, likely through bacterial metabolites.     

In summary, trillions of commensal bacteria inhabit the human gut and they survive on what we feed them. If gut microbiota as a whole is an orchestra, currently only a handful of in­struments (certain bacteria) are heard. Knowing who play the instrument (certain diet or nutrient or other factors) may allow us to gain further insight into diet and microbiota association.  

In the revision, we added one statement in Introduction. Please see page 2. It reads “Our previous analysis shows that a lower score of saturated fatty acid intake in Healthy Eating Index (HEI) was associated with a higher relative abundance of Fusobacterium [21]. This finding suggests that examining the specific nutrients in association with gut microbiota would provide additional insight into the role of diet and microbiota in health. “

Reviewer 2 Report

Dietary pattern analysis examines the effects of overall diet and represents a broader picture of food and nutrient consumption, and maybe more predictive of the effects on microbiota than individual foods or nutrients ( different fatty acids), your data showed those who had a lower intake of SFAs had a significantly higher HEI score and higher intake of total carbohydrates is an indication of that.

In addition, diets (fatty acids also) changes with time, The FFQ obatined 2-3 weeks prior to study might not represent the long term effect of a dietary pattern.

There ought to be a discussion regarding this fact in the discussion or limitation section. 

References

1. Gibiino G, De Siena M, Sbrancia M, Binda C, Sambri V, Gasbarrini A, Fabbri C. Dietary Habits and Gut Microbiota in Healthy Adults: Focusing on the Right Diet. A Systematic Review. Int J Mol Sci. 2021 Jun 23;22(13):6728. doi: 10.3390/ijms22136728. PMID: 34201611; PMCID: PMC8269086.

2. Losno EA, Sieferle K, Perez-Cueto FJA, Ritz C. Vegan Diet and the Gut Microbiota Composition in Healthy Adults. Nutrients. 2021 Jul 13;13(7):2402. doi: 10.3390/nu13072402. PMID: 34371912; PMCID: PMC8308632.

Author Response

Manuscript ID 1760154: Nutrients
Type of manuscript: Article
Title: Dietary fatty acids intake and the colonic gut microbiota in humans

Dear editor and reviewer:

We appreciate reviewers’ insightful comments on our manuscript. Please see the point-by-point response below. Please see the response in brown character. Thank you!

Q1, Dietary pattern analysis examines the effects of overall diet and represents a broader picture of food and nutrient consumption, and maybe more predictive of the effects on microbiota than individual foods or nutrients (different fatty acids), your data showed those who had a lower intake of SFAs had a significantly higher HEI score and higher intake of total carbohydrates is an indication of that.

Response: Thank you for the insightful comment. We have two motivations to conduct the analysis on dietary fatty acids and gut microbiota. First, our analysis was led by our previous finding on HEI and gut microbiota that showed saturated fat may potentially be associated with Fusobacterium. Second, there are numerous studies showing the association between specific fatty acids and various outcomes. We would like to address the question on whether gut microbiota plays any role in such associations. We identified in this study that Sutterella, Fusobacterium, and Tyzzerella was associated with specific dietary fatty acids.

Therefore, we thought it is equally important to uncover specific nutrients in association with gut microbiota to advance our understanding of gut microbiota and diet. The association between Sutterella and fatty acid intake would have been missed if we did not conduct the nutrient-based analysis. 

We added this statement in Introduction: “Our previous analysis shows that a lower score of saturated fatty acid intake in Healthy Eating Index (HEI) was associated with a higher relative abundance of Fusobacterium [21]. This finding suggests that examining the specific nutrients in association with gut microbiota would provide additional insight into the role of diet and microbiota in health.” Please see Page 2. 

Q2. In addition, diets (fatty acids also) changes with time, The FFQ obtained 2-3 weeks prior to study might not represent the long term effect of a dietary pattern. There ought to be a discussion regarding this fact in the discussion or limitation section. 

Response: Thank you for the comment. The questionnaire is designed to ask the food consumption in the past 12 months as we noted in Section 2.2. We included in the limitation on information bias of self-reported diet information in the revision. Please see the 2nd to the last paragraph on Page 10.

Q3. The below references are cited in the manuscript. Thank you for referring.

References

  1. Gibiino G, De Siena M, Sbrancia M, Binda C, Sambri V, Gasbarrini A, Fabbri C. Dietary Habits and Gut Microbiota in Healthy Adults: Focusing on the Right Diet. A Systematic Review. Int J Mol Sci. 2021 Jun 23;22(13):6728. doi: 10.3390/ijms22136728. PMID: 34201611; PMCID: PMC8269086.
  2. Losno EA, Sieferle K, Perez-Cueto FJA, Ritz C. Vegan Diet and the Gut Microbiota Composition in Healthy Adults. Nutrients. 2021 Jul 13;13(7):2402. doi: 10.3390/nu13072402. PMID: 34371912; PMCID: PMC8308632.

Round 2

Reviewer 1 Report

This study examined associations of intake of fatty acids (saturated, trans, monounsaturated, polyunsaturated, n3- and n6- fatty acids) with the structure and composition of the adherent colonic gut microbiota. The issue should arouse the interest of readers. The revised manuscript improved its rationale, provided several methodological details, and added important references and supplementary material.

Authors answered or made comments for all the queries, and modifications in the revised manuscript are very acceptable. Limitations highlight the gaps in knowledge which deserve further investigations.